# Quality Child–Parent Relationships and Their Impact on Intergenerational Learning and Multiplier Effects in Climate Change Education. Are We Bridging the Knowledge–Action Gap?

**Sandra Parth [1],\* , Maximilian Schickl [2],\*, Lars Keller [1],\* and Johann Stoetter [1],\***

1 Institute of Geography, University of Innsbruck, 6020 Innsbruck, Austria
2 Institute of Psychology, University of Innsbruck, 6020 Innsbruck, Austria
\* Correspondence: sandra.parth@uibk.ac.at (S.P.); max@schickl.de (M.S.); lars.keller@uibk.ac.at (L.K.); Hans.Stoetter@uibk.ac.at (J.S.)

**Abstract:** The science–education cooperative venture "Our Common Future: 'eKidZ'—Teach Your Parents Well" explores intergenerational learning processes and the transfer of learning from the younger to the older generation. Students acting as multipliers and their multiplication effect on parents is part of the research setting: 20 high school students, in the role of researchers, investigated the question of whether children who participate in the Climate Change Education (CCE) program "k.i.d.Z.21" passed on their climate-change-related knowledge, attitudes and actions to their parents ($n$ = 91), in comparison to a control group ($n$ = 87). Due to the annual increase in student participants in the CCE project "k.i.d.Z.21" since 2012 ($n$ = 2000), this article can build on the results of a questionnaire regarding the school year 2017/18 ($n$ = 100–120). A Multivariate Analysis of Variance (MANOVA) showed that the "k.i.d.Z.21" project has a multi-faceted knock-on effect on parents, constituting a multiplier effect: increasing knowledge, and, above all, improvements to the child–parent relationship. Additionally, measurable positive effects in the frequency and quality of climate change communication between children and their parents have been observed (Spearman Rank Correlations), but a distinct lack of positive effects regarding changing climate-friendly attitudes or actions have been noted (Pearson Product–Moment Correlation). The importance of the child–parent relationship is a key factor in bridging the knowledge–action gap, and is reviewed in the context of CCE.

**Keywords:** Climate Change Education (CCE); knowledge–action gap; intergenerational learning; multiplier effects; child–parent relationship

## 1. Introduction

Although basic climate change knowledge has been available to the general public for decades [1,2], there is still insufficient action regarding sustainability [3–5]. As such, Climate Change Education (CCE) and Education for Sustainable Development (ESD) programs (dealing with climate-change-related contents and methods) have become essential to promote sustainable changes in individual and climate-friendly community actions [6–11].

As a result, the aim of the CCE project "k.i.d.Z.21" is to increase and strengthen climate change awareness and promote and foster individuals to take necessary against climate change [12]. Positive effects in increasing knowledge and awareness of students have already been observed, but it has been demonstrated that changes in routines to climate-friendly actions are not a natural consequence among the participants [13]. In this context, the so-called "knowledge–action gap" [4,14–16], "attitude–behaviour gap" [17,18] or even the "value–action gap" [19–21] can be used to explain

the phenomenon. However, apart from terminology, the aforementioned terms highlight the gulf that needs to be bridged in order to promote pro-environmental actions and sustainability amongst the wider population [3]. At least this is what knowledge–attitude–behaviour (KAB) models suggest [22] or the knowledge–attitude–action (KAA) model, as it will be referred to here. Therefore, despite many efforts, the transfer of knowledge, and subsequent implementation of knowledge, attitudes, values, etc., into real actions, is yet to be realized. Although sustainable and climate-friendly action is a relatively well-researched topic, the real-world utilization of CCE/ESD information has yet to be satisfactorily implemented. As a result, this paper aims to mitigate the knowledge–action gap by examining the gap in the context of intergenerational learning processes (IGL), which focus on mutual learning between the younger and older generation [23].

According to the slogan, "Teach Your Parents Well", the Climate Change Education project "eKidZ" tries to quantify intergenerational learning and the multiplier effect between children participating at the CCE program "k.i.d.Z.21" and their parents concerning their climate-change-related knowledge, attitudes and actions. In this sense, the multiplier effect is normally prominent. However, more research is needed to determine its impact on the influence of children's parents' climate-friendly decisions [24,25].

A reason for this might be a number of factors, which can either be beneficial or detrimental and influence the intergenerational learning process and the multiplier effect in many ways [26–32]. Another course for the varying characteristics of the intergenerational transfer processes could be different degrees of mutual influence between the generations. A reason for this is that parents often do not perceive the knowledge they acquire from their children, especially with reference to transference of environmental knowledge [33]. Additionally, it has been documented that the transference can influence the parents to different degrees and is affected by the attitudes, awareness and/or general actions of their children and the parents themselves [25,34]. Knafo and Galansky [35] have also stated that more evidence is needed from naturally occurring parent–child interactions.

This point is therefore vital for this study as a contribution of multiplier effects and on bridging the knowledge–action gap. Therefore, we investigate the question of whether the child–parent relationship has a positive effect on the transfer of climate-change-related knowledge, attitudes and actions from the younger to the older generation. Lawson et al. [36] highlighted that future research on intergenerational learning would benefit from in-depth research on variables measuring the strength of the relationships between children and parents (and thus, the frequency and the quality of the communication, which seem to be an important factor in transferring knowledge, attitudes and actions). This study not only contributes to prior research on the knowledge–action gap [4,14–16] by focusing on the child–parent relationship, but also leads to authentic results by deeply involving young people, as the intensive participation of students in the research process itself is a characteristic of transdisciplinary Climate Change Education. Whereas initial insights into the research setting of this project "Our Common Future: 'eKidZ'—Teach Your Parents Well" (2016/2017–2018/2019) have been published [37], this paper aims to investigate the findings and results of students' research regarding intergenerational learning processes and multiplier effects from the younger to the older generation.

## 2. Intergenerational Learning and Multiplier Effects

In contrast to the traditional assumption that the younger generation learns from the older, the concept of intergenerational learning (IGL) today refers to a reversal of the learning process where the "parents also learn from children" [38,39]. In this sense, not only younger people learn from their elders but vice versa. Therefore, learning can be regarded as a common and alternating process that is based on a lifelong approach [40,41]. Especially, intergenerational learning from the younger to the particularly older generation seems to play an important role here [42–45].

The link between intergenerational learning and children's transfer of knowledge, attitudes and/or climate-friendly actions to their parents is themed in several studies, and is often referred to as a multiplier, spillover or catalyst effect [24,31,43,46–48]. In this case, children who participate in an

environmental program and affect their parents in their decisions concerning climate change can be seen as multipliers or catalysts [26,28,32,46,49]. So, the process of transferring knowledge, attitudes and/or actions from children to parents leads to the questions of "what and how" does this multiplier effect occur?

It is already known that there are several influences on the multiplier effect; however, the various effects and their interrelationship is yet to be confirmed [25,28,29,45,49]. Nilsson et al. [25] have noted that the strength of the multiplier effect is dependent on context similarity, personality, self-identity and/or framing, and it has been demonstrated that the link to intergenerational communication and the relationship between children and parents does play an important role [36,46,48,50,51]. According to this complex variety of influencing factors, it is not surprising that increasing knowledge does not necessarily change attitudes or even actions [3,34].

Fortunately, some studies report that students participating in environmental programs do have a positive effect in parents' decisions concerning climate change. In this case, successful intergenerational learning from children is apparent, e.g., general environmental conservation knowledge [52], flood education knowledge [32], waste education actions [53] and energy conservation actions [54]. Lawson et al. [45] advocate the effective and beneficial intergenerational learning effect from children to adults:

> "Environmental education (EE) programs directed at children, but designed with intergenerational learning in mind, also result in the successful transfer of environmental knowledge, attitudes, and behaviours to adults. [ ... ] it is clear that child to adult intergenerational learning is possible, and provides an effective avenue to environmental change that engages both younger and older generation".

This statement is confirmed by their subsequent paper, where their study provides empirical evidence of child-to-parent intergenerational learning associated with climate change concern, especially amongst parents expected to be most resistant. In this context, communication is again connected to intergenerational learning as their findings show increased family discussion around climate change as a key factor in predicting changes in parents' level of concern [36]. So, if a transfer process successfully takes place and a child influences the decisions of adults [55–57], a child or the children can also be called "effective agents" [33]. In summary, the influence of children over their parents is a topic of focus in recent studies, which emphasizes the transfer of knowledge, attitudes and/or actions patterns related to climate change [32,36,43,45,58]. In this case, Gallagher and Fitzpatrick [48] even refer to intergenerational learning as a "win–win situation" in creating opportunities for developing relationships and positive opinions. Overall, the success of intergenerational learning depends on the willingness to learn from each other and how children's opinions are valuated in their family [43]. Lawson et al. [36] highlighted that the strength of relationships between children and parents could also be essential for intergenerational learning and therefore affect multiplier effects.

## 3. Quality Child–Parent Relationship in the Context of Bridging the Knowledge–Action Gap

Why is there a difference between what people say and what they do? This is a broad and multi-layered question. Early studies attempted to explain the relationship between knowledge, attitudes and actions, and many presented their findings in a linear model. The idea is that environmental knowledge leads to environmental attitudes, which in turn was thought to lead to pro-environmental action. If this linear process was true, educating people about environmental issues would automatically result in pro-environmental actions [3]. However, this fails to account for that fact that our actions do not always occur as a result of rational thinking. Moreover, these linear connections are multivariable, and a slight change to one factor can have unexpected consequences on the other [59]. Numerous research fields (economics, psychology, education, tourism, environmental sciences, etc.) have tried to understand or even influence this cause and effect relationship in the psychology of human action. Darnton [22] attempted to clarify our understanding by creating action models. However, it has been demonstrated that too many different networked variables can influence

actions; these include knowledge, habits, values, attitudes, beliefs, emotions, ethical norms, and identity. This list could be expanded to include numerous internal factors, such as interest and self-efficacy experiences, as well as external factors such as social environment and political and institutional conditions, which also influence everyday decisions and actions [60].

Due to its complexity and the diffuse effect of internal and external factors on human action, focus on the relationship between children and their parents should bring a new perspective and understanding to the discussion on intergenerational learning and the transfer of knowledge, attitudes and/or actions [31,43]. Williams et al. [32] suggest family relationships as a key factor for successful intergenerational learning. Conversely, intergenerational learning can also be seen as a means of emphasizing the relations between generations [61].

Therefore, all variables should be viewed as a network in our mind, rather than the aforementioned linear KAB model. This new proposed cross-linked KAA model, expanded to include the child–parent relationship variable, is presented in Figure 1.

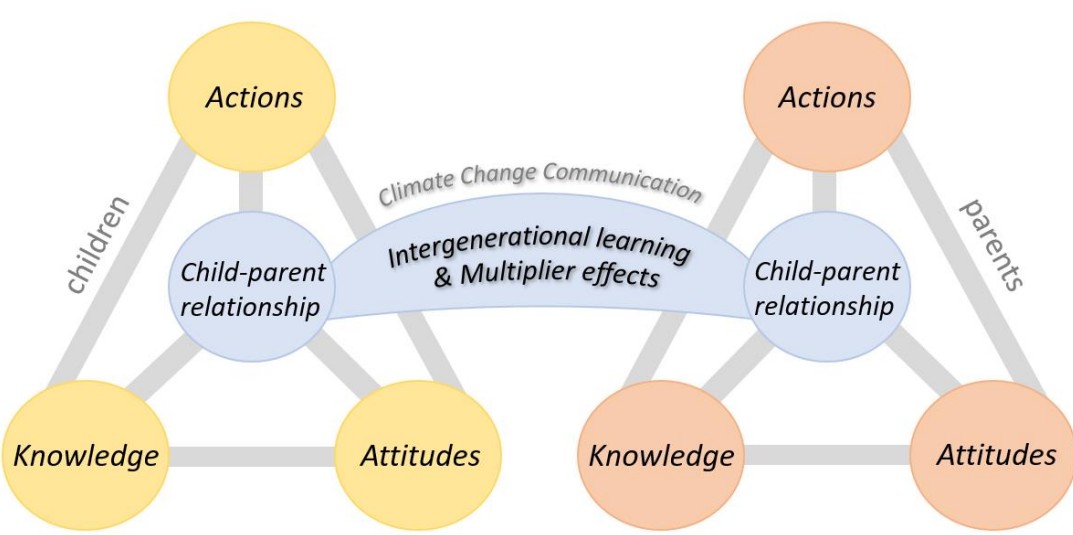

**Figure 1.** A cross-linked knowledge–attitude–action (KAA) model depicting how the personal relationship to knowledge, attitudes and actions and the intergenerational learning between a child and parent functions (own graph).

In this cross-linked KAA model, children as well as parents are presented with their own mental perceptions, associations and concepts about knowledge, attitudes and/or actions regarding climate change. The triangle-shape demonstrates that all variables are linked together. As climate-friendly actions are a crucial aspect of sustainability, they are located at the top. Moreover, the child–parent relationship is located at the center of this model, and all of the other variables are linked to it, as it plays the dominant role in this study. According to this model, the transfer of knowledge, attitudes and/or actions between generations occurs via communication. Note, this might lead to intergenerational learning and multiplier effects. This model compliments Knafo and Galansky's [35] focus on further research in child–parent relationship.

Communication plays an integral part in the child–parent relationship, and should therefore be considered in our debate on bridging the knowledge–action gap. Indeed, Lawson et al. [36] provided valuable insights, noting that, "[their] findings show increased family discussion around climate change was a key factor in predicting changes in parents' concern levels". The authors also point out that communicating climate concerns between children and parents successfully may reflect the robustness of the child–parent relationship to any socio-ideological threats that adults may typically associate with climate change. This indicates that the quality of the relationship and the communication between children and their parents are important for making changes. Thus, children may provide a communication pathway that is resilient to longstanding socio-ideological barriers to learning, caring

and ultimately acting on climate change [36]. Quality intergenerational communication is a reciprocal process, meaning that parents have to be willing to learn from their children and let them participate in decisions on climate-change-related topics. Indeed, this condition must be met in order to bridge the gap between knowledge and action [46,62].

## 4. Method

### 4.1. Research Design of the CCE Projects "k.i.d.Z.21" and "eKidZ"

The CCE project "'k.i.d.Z.21'—Competent Into The Future" [12] started as a joint venture between Karl-von-Closen (KvC) high school (Germany) and the University of Innsbruck (Austria). Since the initiation in 2012, around 120 students have participated in annual seminars, research weeks and questionnaires. From 2012 to 2020, the project has slowly grown to 700 participants, and following the involvement of a greater number of Austrian high schools in 2015, the participant number has increased to 2000. The outstanding cooperation of all partners involved and the long-term cooperation between the University and the various schools has led to the relative sample size being large enough to enable quantitative research into numerous long-term multiplier effects, and document their influence on the intergenerational learning process. Grasping this opportunity and with reference to CCE, 20 students from different 8th grade classes in KvC accompanied by scientists from the University of Innsbruck were selected to take part in "Our Common Future. 'eKidZ': Teach Your Parents Well" project for intergenerational learning and multiplier effect research. Figure 2 gives an overview on the research design of the two closely connected CCE projects: "k.i.d.Z.21" and "eKidZ".

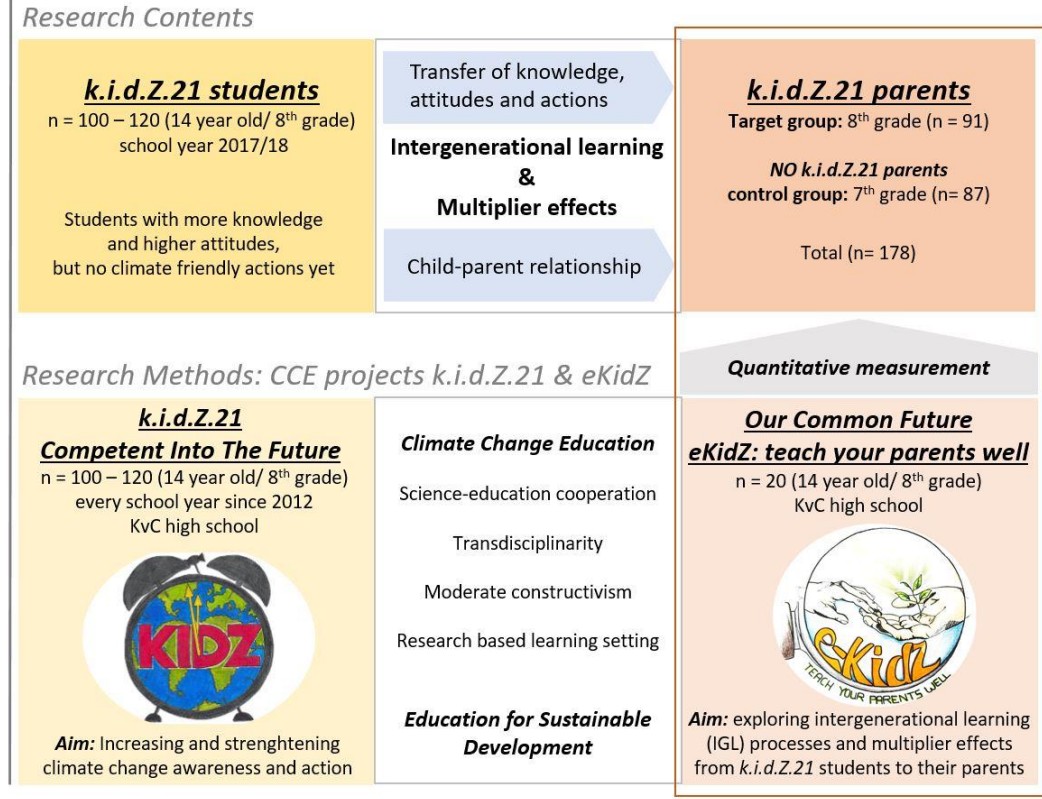

**Figure 2.** Research design of the CCE projects "k.i.d.Z.21" and "eKidZ" (own graph).

Both projects, "k.i.d.Z.21" and "eKidZ", are based on theories related to CCE and ESD, e.g., science-education cooperation, transdisciplinarity, moderate constructivism and research-based learning settings [12,37]. In "k.i.d.Z.21", students gain more knowledge and higher attitudes in various educational modules, but climate-friendly actions are not yet measurable. These education

modules are beyond the traditional curriculum of the school program (see Appendix A) and are further described and evaluated in Keller et al. [13].

In "eKidZ", students and scientists complete research together—students generate their own research questions and methods, and scientists foster a dialogue for developing a common questionnaire on the topics of climate-related knowledge, attitudes, actions and child–parent relationship [37]. To preserve the authenticity of the students' scientific work, the questionnaire was subjected as much as necessary to a further scientific selection procedure.

## 4.2. Data Collection

A two-group experimental design was setup. The groups were divided between parents whose children had participated in the Climate Change Education project "k.i.d.Z.21" (=8th grade/*n* = 91) as an experimental group and parents whose children had not yet participated in the Climate Change Education project "k.i.d.Z.21" (=7th grade/*n* = 87) as the control group. Parents who had children in both classes (7th and 8th grade) and parents of students participating in the research project "eKidZ" were excluded. Pre-test pencil–paper interviews took place between September and October 2017, with a follow-up taking place between July and October 2018. Parents who participated in both interviews remain in the dataset, with a total of *N* = 178 cases. After data collection, the paper questionnaires were entered into SPSS by the students and the scientists.

## 4.3. Data Analysis

First, the students used their questionnaire items to create the four factors of climate-related knowledge, attitudes, actions and child–parent relationship for the statistical program SPSS. As the students also had to decide on their own research process, no other scientific standards were applied to this scientific procedure.

The knowledge factor includes items such as "How do you assess your knowledge about climate change" and "How strongly are the following six phenomena influenced by climate change? Melting of the polar caps—environmental refugees—shifting of the seasons—increasing flood events—deforestation of the rainforest (neg.)—tidal change (neg.)".

The factor attitudes mainly consists of the willingness to behave in a climate-friendly manner, e.g., to pay taxes on petrol or diesel-powered vehicles or to do without heating or electricity, as well the belief that others, such as politicians, are responsible for it or the conviction to be able to do something about climate change.

The factor actions is divided into mobility and consumption. Here, the variable consumption is multi-faceted: from the purchase of electrical equipment to eating habits and living in climate change friendly housing. So, besides general consumption action, a few variables are considered, namely, food and housing.

The factor child–parent relationship is based on the following questions:

- How often do you talk to your child/children about climate change?
- How do you rate the quality of these discussions?
- How much do you agree with these statements [these changes] in relation to these talks?
- In which climate-impacting decisions does your child/children influence you?

A student came up with the idea to use the calculation method of a two-way repeated-measures multivariate analysis of variance (MANOVA) in order to reduce the family-wise-error-rate (FWER) to 5 percent. The FWER is the probability of getting at least one false-positive finding within the whole study. The MANOVA helps to control this type of error. So, after the factors were formed, the statistical procedure MANOVA was applied to all five factors as dependent variables and the group (experimental/control) as independent variable.

According to Bortz and Döring [63], interactions can only occur in experiments with two or more independent variables. A significant interaction effect in analysis of variance means that both factors

cannot be added together, but interact in a different way. This refers to an interaction where the effect of one of the two variables is dependent on the effect of the other variable. The MANOVA showed an interaction effect between the time of measurement and the group (school grade), $F(5,173) = 3.36$, $p = 0.006$. However, there are no effects within the groups, $F(5,173) = 2.06$, $p = 0.073$, or between the groups, $F(5,173) = 0.95$, $p = 0.451$. Therefore, the interaction is discussed in more detail.

*4.4. Findings*

The factors knowledge ($r = 0.24$) and child–parent relationship ($r = 0.19$) show a small interaction effect [64], whereas in attitudes and actions no significant effects are measurable. Table 1 shows the interactions between the time of measurement and the group (school grade). The within-group effects (time of measurement) and between-group effects (school grade) are not listed here, since they were not detected by the MANOVA before.

**Table 1.** Interactions showed by five subsequent two-way repeated measures ANOVAs.

| Factor | F | p | r |
|---|---|---|---|
| Knowledge | 10.89 | 0.001 * | 0.24 |
| Attitudes | 0.42 | 0.517 | |
| Action (Mobility) | 0.01 | 0.921 | |
| Action (Consumption) | 0.08 | 0.770 | |
| Child–Parent Relationship | 6.78 | 0.010 * | 0.19 |

\* $p \leq 0.05$. *df1* = 1. *df2* = 177. *r* = effect size of interaction derived from *F*-statistics.

*4.5. Graphs*

For graph interpretation, it is helpful to assess the following indications: if there is no interaction and the factors "only" interact additively, the graphs shown in the interaction diagram are parallel. The more they deviate from the parallelism, the more likely this is to indicate the existence of an interaction effect [63]. In order to exclude coincidences in this context, a confidence interval (CI) can also be used. If the center of the CI were the true mean of the population, 95 percent of the experiments would have the measured mean within the 95% CI. The two lines of the graph are offset for reasons of clarity. The time of measurement is nevertheless the same.

4.5.1. Knowledge

In Figure 3, the result formulated in the hypothesis can be seen very clearly. The knowledge of the parents of the 8th grade students has increased above the knowledge of the parents of the 7th grade students.

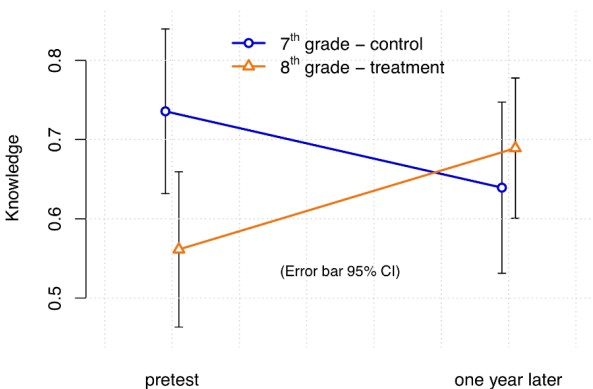

**Figure 3.** Students effect on parents knowledge.

This effect can be attributed to the "k.i.d.Z.21" project, as there is an interaction between school grade and time of questioning on knowledge, $F(1,177) = 10.89$, $p = 0.001$. The "k.i.d.Z.21" project has a positive effect on the knowledge of the parents of the respective year-group. The initial MANOVA showed neither an effect of the measurement time alone nor of the grade level alone.

### 4.5.2. Attitudes

Figure 4 shows that the parents of the 8th graders in the first round of the survey have the same attitudes towards whether and how to behave in a climate-friendly manner than the 7th graders. In the second round of the survey, both grades have retained the same attitudes.

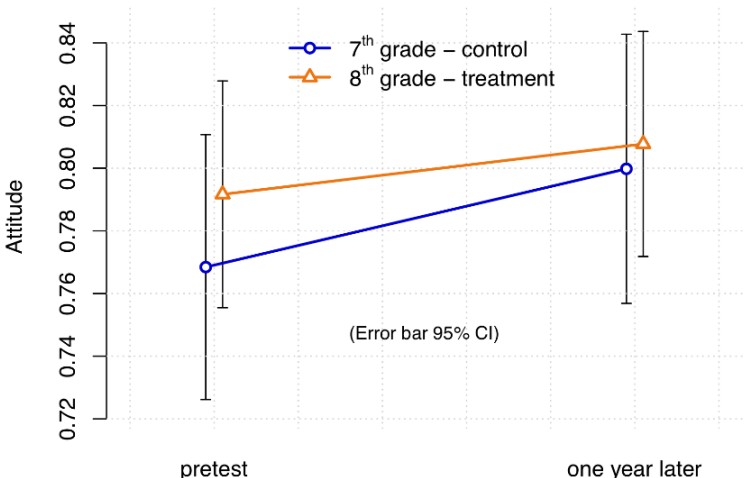

**Figure 4.** Students effect on parents attitudes.

The initial MANOVA showed no between-subject and within-subject effects. This also applies to the attitudes. A two-way repeated measures ANOVA showed no interaction effect between time and group, $F(1,177) = 0.42$, $p = 0.517$. In summary, this means that the attitudes of the parents have not changed either over time nor by the "k.i.d.Z.21" project, and do not differ between 7th and 8th grade parents.

### 4.5.3. Actions

The project divides climate-friendly actions into the four categories: nutrition, consumption, mobility and housing. In the evaluation, the students decided on mobility and consumption. For this reason, the two categories nutrition and housing are not mentioned here, but mobility and consumption are represented in Figures 5 and 6.

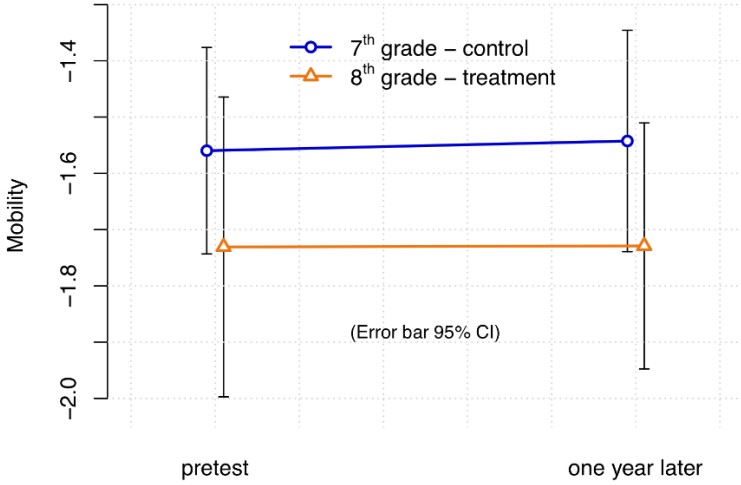

**Figure 5.** Students effect on parents action (mobility).

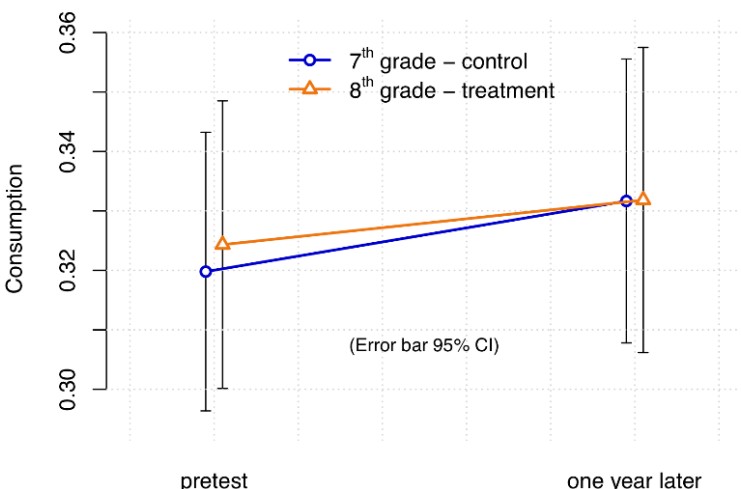

**Figure 6.** Students effect on parents action (consumption).

A two-way repeated measures ANOVA shows no interaction effect for mobility, $F(1,177) = 0.01$, $p = 0.921$, or in consumption, $F(1,177) = 0.09$, $p = 0.770$. Within-subject effects and between-subject effects are negotiated by the initial MANOVA. Thus, in both cases the actions of the parents have not changed either over time or by the "k.i.d.Z.21" project and do not differ between 7th and 8th grade parents.

### 4.5.4. Child–Parent Relationship

As the child–parent relationship plays an important role in transferring knowledge, attitudes and/or action, Figure 7 might be of interest in connection to intergenerational learning and the multiplier effect.

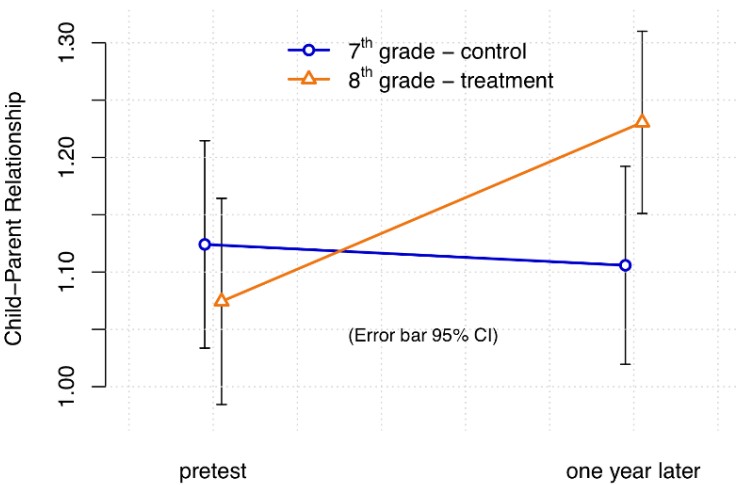

**Figure 7.** Students effect on child-parent relationship.

This graph of the child–parent relationship is similar to the graph of the knowledge as the interaction can be seen very clearly. It is a picture-book like graph for an experimental effect. The relationship between the children of the 8th grade and their parents has increased beyond the relationship between the children of the 7th grade and their parents. A two-way repeated measures ANOVA shows an interaction between school grade and time of questioning on child–parent relationship, $F(1,177) = 6.78$, $p = 0.010$. This can be attributed to the "k.i.d.Z.21" project. The "k.i.d.Z.21" project has a positive effect on the child–parent relationship of the parents of the respective year-group. As the initial MANOVA found, there is neither an effect of the measurement time alone on the child–parent relationship, nor an effect of the grade level alone.

*4.6. Additional Null Hypothesis Testing (NHST)*

Correlations between the factors of knowledge, attitudes, actions (mobility and consumption) and child–parent relationship are tested and listed in Table 2.

**Table 2.** Correlations between all factors.

|  | 1 | 2 | 3 | 4 |
|---|---|---|---|---|
| (1) Knowledge |  |  |  |  |
| (2) Attitudes | 0.28 */0.37 * |  |  |  |
| (3) Child–Parent Relationship | 0.01/0.27 * | 0.12/0.13 |  |  |
| (4) Action (Mobility) | 0.09/0.07 | 0.17/0.13 | 0.01/0.02 |  |
| (5) Action (Consumption) | −0.02/0.05 | 0.05/0.14 | 0.11/0.05 | −0.08/0.09 |

*Note.* * = $p < 0.005$. Bonferroni corrected level of significance = 0.00025, to achieve a FWER of 0.5% over the whole table. Values are product–moment–correlation coefficients $r$. Values for both measurements are shown: pretest/one year later.

The two significant correlations for the one-year-later measurement are listed below.

(1)　A Pearson product–moment correlation shows an effect between knowledge and attitudes, $r(177) = 0.38$, $p < 0.001$. The more knowledge one has about climate change, the more willing one is to do something about it.

(2)　A Pearson product–moment correlation shows an effect between knowledge and child–parent relationship, $r(177) = 0.28$, $p < 0.001$. The greater the knowledge a child has about climate change and the better the child–parent relationship, the more the child has influence in climate-change-related communications, such as decisions in consumption, mobility, nutrition and living.

Product–moment correlations between frequency of conversations and their quality showed following effects:

7th grade, pretest: $r(84) = 0.28$, $p = 0.008$.
7th grade, one-year-later: $r(81) = 0.31$, $p = 0.004$.
8th grade, pretest: $r(87) = 0.47$, $p < 0.001$.
8th grade, one-year-later: $r(88) = 0.51$, $p < 0.001$.

In summary, in the 7th and 8th grades, the quality of the conversations increases with frequency.

## 5. Discussion

This experiment as a longitudinal study shows to what extent the CCE project "k.i.d.Z.21" induces intergenerational learning and multiplier effects from children to their parents in comparison to the control group.

### 5.1. With Reference to the Question 'What' Do Students Multiply to Their Parents

The results illustrate that students in the CCE project, "k.i.d.Z.21" has positive effects on the climate-change-related knowledge of their parents *[F(1,177) = 10.89, p = 0.001]* but no significant results concerning the change of attitudes *[F(1,177) = 0.42, p = 0.517]* and climate-friendly actions *[mobility, F(1,177) = 0.01, p = 0.921, /consumption, F(1,177) = 0.09, p = 0.770]*, neither in the respective group nor in the control group.

The general assumption that the transfer of knowledge, attitudes and/or actions are a linear occurrence [22,65] can only be approximated by this study. This means that knowledge about climate change alone does not necessarily culminate in climate action, which indicates there is a gap between knowledge and action but also highlights that the chain of effects is not as linear as previously thought [34,59]. So, the correlation, "the higher the knowledge about climate change, the higher the willingness to do something about it "*[r(177) = 0.38, p < 0.001]* suggests some linearity between knowledge and attitudes and might be therefore helpful in bridging the knowledge–action gap. However, in our study evidence for changing action towards sustainability is still missing, for both children and parents. This finding could be of interest to further research, as other studies have proven that there is a positive learning effect between children and their parents' actions [46,53,54]. Therefore, the fact that this paper was able to monitor a positive knowledge transfer but not attitudes or actions, while the aforementioned publications could, highlights we must first improve our understanding on "how" multiplier effects work.

### 5.2. "How" Students Multiply to Their Parents

The hypothesis that the child–parent relationship is a key factor in intergenerational learning processes can be confirmed by these findings. The project "k.i.d.Z.21" shows positive effects on the relationship between children and their parents *[F(1,177) = 6.78, p = 0.010]*. This means that the quality of the relationship between children and parents is present in multiplier effects, especially in the transfer of knowledge and can be statistically proven, thus confirming Lawson et al.'s [36,45] assumptions. This is due to climate change communication, because a positive correlation between the frequency and the fact that the quality of climate-change-related discussions is measurable. Our results underline the findings of Lawson et al. [36], and show that increased family discussion around climate change is a key factor in predicting changes in parents' concern levels ($p = 0.048$; Cohen's $f^2 = 0.36$). So, the assumption that the child–parent relationship is a key factor in intergenerational learning processes can be affirmed, but is it a key factor for bridging the knowledge–action gap as well?

### 5.3. Child–Parent Relationship for Bridging the Knowledge–Action Gap?

The project "eKidZ" has highlighted positive correlations between the child–parent relationship and the transfer of knowledge: "the greater the knowledge a child has about climate change and

the better the child–parent relationship, the more the child has influence in climate-change-related communications, such as decisions in consumption, mobility, nutrition and living" *[r(177) = 0.28, p < 0.001].*

In this case, the correlation between intergenerational learning and the child–parent relation as proposed by Brownell and Resnick [61] and Williams et al. [32] fit together. According to this finding, the child–parent relationship factor does influence the transfer of knowledge. Although a significant change in attitudes and actions has not yet been measured, the correlation between increasing knowledge and child–parent relationship seems to be a fruitful way to contribute to the research of bridging the knowledge–action gap. Even though the "gap" in this study was unable to be closed, general evidence of children influencing their parents in their climate-friendly action due to family communications has already been provided [36,45]. Hopkins [46], for instance, confirms action changes due to intergenerational learning: "It is likely that a number of action changes in family resulted directly from the amount of interpersonal communication that took place after the intervention". So, the proposed cross-linked model (KAA) with the additional factor of child–parent relationship might be helpful in further CCE programs in which research is conducted with students, especially those that involve intergenerational learning and multiplier effects for bridging the knowledge–action gap.

The research in this report therefore supports and confirms Knafo and Galansky's [35] desire for having more evidence of the factor child–parent relationship.

### 5.4. Further Learning Effects in CCE Projects "k.i.d.Z.21" and "eKidZ"

The results of the students' study produced an interesting discovery, because stimulating intergenerational learning and multiplier effects were not originally the main objective of "k.i.d.Z.21". Even then, students participating at the CCE program "k.i.d.Z.21" with positive effects in knowledge and attitudes [13] did not transfer attitudes and actions to their parents. In this case, the initial working hypothesis of the "eKidZ" students—"Changes in action are to be expected due to possible gains in knowledge"—can be reconsidered critically but also seen as a positive vision with an empowered young generation in the future. In order to support this statement and to give young people a voice, we can confirm that students participating in CCE projects like "k.i.d.Z.21" can be seen as "effective agents" [33].

Riede et al. [66] made a recommendation regarding CCE in the context of climate change communication: "Especially, the fact that students are not instructed with knowledge and persuaded to change their attitudes and/or actions, but instead encouraged to see climate change through their very own eyes and come up with their very own ideas to tackle climate change challenges, ensures a greater acceptance and effectivity among the target group". This report is therefore an appeal to CCE programs to let students become active learners and to empower them in their climate-friendly actions, as it is the case in "k.i.d.Z.21" or as equal researchers, like in "eKidZ". Apart from this, strengthening and empowering young people in taking their own actions and communicating their climate change ideas in family discussions is part of CCE/ESD. The following statement from an "eKidZ"—student shows the importance of climate-friendly actions in a young person's mind: "More important than knowledge, however, is the subsequent action". This can be seen as another achievement of the science-education cooperation "eKidZ", whereby raising climate change awareness was not a main goal of "eKidZ". Both projects, "k.i.d.Z.21" and "eKidZ", were able to record successes that were achieved either directly or indirectly. Therefore, further research could investigate the question of how this research setting fosters intergenerational learning and multiplier effects.

In summary, with regards to "what and how" multiplier affects children to adult learning, this paper can conclude that parents mainly gain knowledge from their children in climate-change-related discussions, whereby the quality of the child–parent relationship positively influences the success of intergenerational learning effects. In the end, the transfer of knowledge, attitudes and/or actions mainly depends on one's own concepts and the quality of relationship between children and parents. This suggests that a CCE/ESD project should also consider the following reflective statement, "It is

possible for children and parents to act independently of one another's attitudes and behaviours" [46], which means that a deeper look into a research setting based on CCE/ESD is needed.

**Author Contributions:** Conceptualization, S.P., L.K. and M.S.; methodology, S.P. and M.S.; software, M.S.; validation, S.P. and M.S.; formal analysis, S.P. and M.S.; investigation, S.P. and J.S.; resources, J.S.; data curation, S.P. and M.S.; writing—original draft preparation, S.P., M.S. and L.K.; writing—review and editing, S.P., M.S. and L.K.; visualization, S.P. and M.S.; supervision, L.K., and J.S.; project administration, L.K., and J.S.; funding acquisition, L.K., and J.S. All authors have read and agreed to the published version of the manuscript.

**Funding:** This research was funded by Robert Bosch Stiftung GmbH.

**Acknowledgments:** The authors would like to thank the Robert Bosch Stiftung GmbH for their financial support of the study and the Karl-von-Closen Gymnasium for their longstanding cooperation. We also sincerely thank our 20 students who participated at the "eKidZ" project: Maximilian Bauer, Michael Bösl, Sophia Breu, Maximilian Brummer, Sebastian Buchauer, Jonas Dietrich, Tamara Dömötör, Simon Erber, Lena Girgnhuber, Antonia Goller, Anna Grabmaier, Chiara Laubenbacher, Alexander Moosner, Celine Pollner, Emma Riedler, Alex Steckermaier, Anna-Lena Thölstede, Julia Wagner, Thomas Wimmer, Florian Zellmer, and the project teachers for supporting "eKidZ" for 3 years: Birgit Danner, Simon Schiller and Heinz Hemberger. We would also like to thank Christopher Standley, for his language assistance and proofreading of this article. This manuscript benefited from the thoughtful comments from reviewers and we thank them for their time.

**Conflicts of Interest:** The authors declare no conflict of interest.

## Appendix A

The linking of the Bavarian curriculum of class 8 in the subject Geography: http://www.gym8-lehrplan.bayern.de/contentserv/3.1.neu/g8.de/id_26283.html (accessed on 20 August 2020).

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
