# Peer review of "Quality Child–Parent Relationships and Their Impact on Intergenerational Learning and Multiplier Effects in Climate Change Education. Are We Bridging the Knowledge–Action Gap?"

_sustainability, doi:10.3390/su12177030_

Round 1

Reviewer 1 Report

 The additions add to the conversation on IGL in climate change context settings. 

Author Response

Dear reviewer,

Thank you very much for this positive comment. I'm looking forward to further cooperation.

Reviewer 2 Report

It could be considered that the discussion and conclusions could be elaborated a little more, on the other hand there are quotations missing in the bibliography:

41.- (Papaoikonomou, Ryan & Ginieis 2011; Langenbach et. al. 2019)

42.- (Blake 1999; Flynn, Bellaby & Ricci 2010; Chung & Leung 2007)

68.- Paco 2017 = Paço, 201

424.- Riede et. al.

Finally, the number of authors in the citations could be increased, so as not to repeat so many of them. There are authors whose quotations are more than four times.

Taking into account these considerations, the article can be taken to a higher level, although the current one is good.

Author Response

Dear reviewer,

thank you very much for this helpful comment. I did following approvements:

Added missing quotations in the bibliography

  • Papaoikonomou, E.; Ryan, G. & M. Ginieis (2011): Towards a Holistic Approach of the Attitude Behaviour Gap in Ethical Consumer Behaviours: Empirical Evidence from Spain. In: Int Adv Econ Res 17, 2011 (1), S. 77–88.
  • Langenbach, B. P.; S. Berger; T. Baumgartner & D. Knoch (2020): Cognitive Resources Moderate the Relationship Between Pro-Environmental Attitudes and Green Behavior. In: Environment and Behavior 52, 2020 (9), S. 979–995.
  • Blake, J. (1999): Overcoming the ‘value‐action gap’ in environmental policy: Tensions between national policy and local experience. In: Local Environment 4, 1999 (3), S. 257–278.
  • Flynn, R.; P. Bellaby & M. Ricci (2009): The value-action gap in public attitudes towards sustainable energy: the case of hydrogen energy. In: The Sociological Review 57(2_suppl), 2009, S. 159–180.
  • Chung, S.S. & M.M-Y. Leung (2007): The value-action gap in waste recycling: the case of undergraduates in Hong Kong. In: Environmental management 40, 2007 (4), S. 603–612.

Tried to avoid repetition of authors by replacing them with other authors or cut them out

  • Please make sure that my change is appropriate and improves the quality
  • New authors:
    • Chang 2014 (line 32)
    • Bottery 2016 (line 48)
    • Watts 2017 (line 82)
    • Boström 2017 (line 84)
    • Cocco-klein & Mauger 2018 (line 130)
    • Dikcius et al 2016 (line 130)

Added Doi Link in the bibliography where possible

Added authors contribution, funding, acknowledgements, no conflict of interests, appendix A and B

Reviewer 3 Report

  • Some references to curriculum of geography in Austria for 8th grade should be included to contextualize better the knowledge contents of students
  • Powerful knowledge on geography education literature would also be cited in the articule
  • Another important title is missing https://www.routledge.com/Climate-Change-Education-Knowing-doing-and-being-1st-Edition/Chew-Hung/p/book/9780415641968

Author Response

Dear reviewer,

thank you very much for your advice. I tried to improve the quality of the article:

Added the important title about CCE in the introduction

C.H., Chang (2014): Climate Change Education. Knowing, doing and being. Hoboken: Taylor and Francis. Online under: http://gbv.eblib.com/patron/FullRecord.aspx?p=1687392.) (line 32)

Mentioned curricula of Geography in appendix A

In order to better contextualize the knowledge content on climate change of the students participating in the CCE program k.i.d.Z.21, the linking of the Bavarian curriculum of class 8 in the subject Geography is mentioned in appendix A: http://www.gym8-lehrplan.bayern.de/contentserv/3.1.neu/g8.de/id_26283.html [20.08.2020]

-> But not sure if this is helpful as the curricula is in the German language

Changed citation stile in the bibliography

BEFORE:

Ballantyne, R.; S. Connell & J. Fien (2006): Students as catalysts of environmental change: a framework for researching intergenerational influence through environmental education. In: Environmental Education Research 12, (3-4), S. 413-427.

AFTER:

Ballantyne, R.; S. Connell & J. Fien. Students as catalysts of environmental change: a framework for researching intergenerational influence through environmental education. Environmental Education Research 12, 2006, (3-4), 413-427.

  • Please let me know if this stile is appropriate

kind regards